# Self-Adaptive Teacher-Student framework for colon polyp segmentation from unannotated private data with public annotated datasets

**Yiwen Jia, Guangming Feng, Tang Yang, Siyuan Chen, Fu Dai** ⓘ *

Department of Gastroenterology, The Third Affiliated Hospital of Anhui Medical University, Hefei, Anhui, China

* swtldjqw@sina.com

## Abstract

Colon polyps have become a focal point of research due to their heightened potential to develop into appendiceal cancer, which has the highest mortality rate globally. Although numerous colon polyp segmentation methods have been developed using public polyp datasets, they tend to underperform on private datasets due to inconsistencies in data distribution and the difficulty of fine-tuning without annotations. In this paper, we propose a Self-Adaptive Teacher-Student (SATS) framework to segment colon polyps from unannotated private data by utilizing multiple publicly annotated datasets. The SATS trains multiple teacher networks on public datasets and then generates pseudo-labels on private data to assist in training a student network. To enhance the reliability of the pseudo-labels from the teacher networks, the SATS includes a newly proposed Uncertainty and Distance Fusion (UDFusion) strategy. UDFusion dynamically adjusts the pseudo-label weights based on a novel reconstruction similarity measure, innovatively bridging the gap between private and public data distributions. To ensure accurate identification and segmentation of colon polyps, the SATS also incorporates a Granular Attention Network (GANet) architecture for both teacher and student networks. GANet first identifies polyps roughly from a global perspective by encoding long-range anatomical dependencies and then refines this identification to remove false-positive areas through multi-scale background-foreground attention. The SATS framework was validated using three public datasets and one private dataset, achieving 76.30% on IoU, 86.00% on Recall, and 7.01 pixels on HD. These results outperform the existing five methods, indicating the effectiveness of this approach for colon polyp segmentation.

## Introduction

Colon polyps have become a focal point of research due to their heightened potential to develop into appendiceal cancer, which is among the cancers with the highest mortality rate globally. Particularly, 30% of adults over the age of 50 suffer from colon polyps [1]. Though largely benign, a fraction can evolve into colon cancer—a notably fatal condition, especially

ⓘ OPEN ACCESS

**Data Availability Statement:** We provide part of the data for validation, which is available at https://osf.io/nk9py/files/osfstorage/66469e70e8eec5742a6bec35.

**Funding:** This work is supported by the research funding of Anhui Medical University (2022xkj105).

when diagnosed late. Colorectal cancer is the third most deadly and fourth most commonly diagnosed cancer in the world. Nearly 2 million new cases and about 1 million deaths are expected in 2018 [2]. The main preventative measure for colon polyps is early detection through a colonoscopy, an exam performed by endoscopists using a colonoscope—a device equipped with a light and a small camera—to inspect the gastrointestinal tract and potentially remove any polyps identified. However, the procedure has a significant drawback; it overlooks 26% of polyps [3], a factor contributing to late diagnosis and poorer outcomes. This oversight largely arises due to the varying appearances of polyps, including differences in size, color, and structure, as illustrated in Fig 1, with smaller polyps posing a particular challenge due to their low contrast against the normal lining of the colon and rectum.

Colon polyp segmentation based on deep learning (DL) can enhance diagnostic accuracy and consistency through automated analysis, which reduces human errors and discrepancies. Firstly, colon polyp segmentation significantly speeds up the diagnostic process. DL-based colon polyp segmentation can quickly analyze large sets of medical images to identify and segment polyps, reducing the time healthcare professionals need to spend on this task, which not only expedites diagnosis but also ensures that treatment can commence promptly, potentially leading to better health outcomes. Secondly, colon polyp segmentation assists in tailoring personalized treatment plans. By providing detailed information about the morphology and size of individual polyps, the segmentation allows healthcare providers to develop targeted treatment strategies. For instance, a patient with small, non-aggressive polyps might receive a different, perhaps more conservative, treatment plan compared to someone with large, irregularly shaped polyps exhibiting malignancy characteristics. Lastly, colon polyp segmentation can aid in the identification of small or easily overlooked lesions. DL-based colon polyp segmentation models can be trained to identify subtle signs and patterns that might indicate the presence of polyps, even in difficult-to-interpret images. This enhanced detection capability can lead to early and more accurate diagnoses, providing the opportunity for interventions that prevent potential progression to more severe conditions.

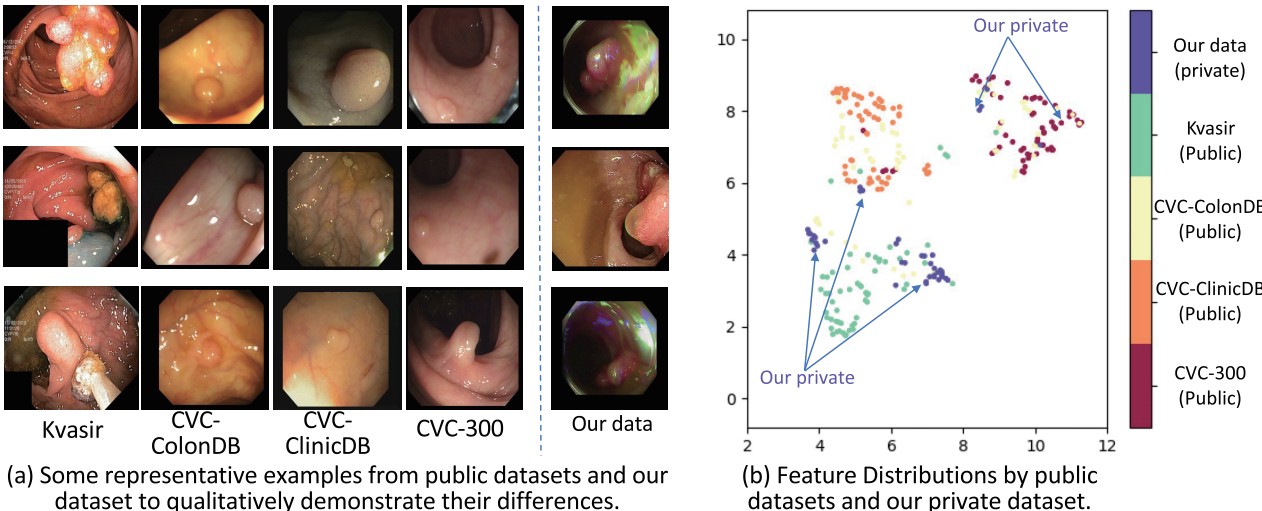

(a) Some representative examples from public datasets and our dataset to qualitatively demonstrate their differences.

(b) Feature Distributions by public datasets and our private dataset.

**Fig 1. The distributions of public datasets are different from the distribution of our private data.** The left illustrates examples from public datasets and our private data to show the difference in image content and light conditions. The right illustrates the feature distribution of different data sets in low-dimensional space by using the UMAP algorithm [4]. The two axes represent the two main dimensions after dimensionality, which are new coordinate systems generated through the UMAP optimization process based on the geometric structure of the data distribution instead of corresponding to specific features in the original feature space.

Although numerous colon polyp segmentation methods have been developed using public colon polyp datasets, they tend to underperform on private datasets due to inconsistencies in data distribution and the difficulty of fine-tuning them on unannotated private datasets. Specifically, Fig 1 illustrates the distribution differences between public colon polyp datasets and our private dataset. From the representative examples in Fig 1(a), it is evident that the lighting and color in our dataset differ from those in public datasets. For instance, our dataset contains green or blue bright spots, which are not observable in public datasets. Additionally, there are unidentified liquids in our dataset due to the different stages of colon polyps. The feature distribution in Fig 1(b) further illustrates that the distribution of our private dataset differs from public datasets. Moreover, private colon polyp datasets generally lack annotations, making it difficult to fine-tune existing methods on private datasets. Annotation requires experts in the relevant field to identify and label specific structures or features in the images, which is a meticulous and technically demanding task. Therefore, the lack of supervision in private data without annotations hinders the fine-tuning of methods developed on public datasets.

This paper proposes a Self-Adaptive Teacher-Student framework (SATS) to segment colon polyps from unannotated private data by leveraging multiple public polyp datasets. The SATS trains multiple teacher networks separately on multiple public polyp datasets. These teacher networks predict segmentation masks on the private data, which are used as pseudo-labels to train a student network on the private dataset. To build more reliable pseudo-labels, the SATS adaptively determines the weights of the multiple predicted masks using a novel self-supervised similarity evaluation strategy. This strategy assigns a larger weight to the teacher network trained with the public dataset that is closest to the input private image. Additionally, to enhance the accuracy of the segmentation results, the SATS constructs a novel coarse-to-fine network architecture. This architecture simulates the clinicians' process of identifying polyps first and then progressively refining the coarse predictions by focusing on ambiguous areas. Our contributions are summarized as follows:

- We propose a new transfer learning protocol to achieve colon polyp segmentation on an unannotated private dataset. The protocol efficiently leverages the robust learning capabilities of public datasets to parse and understand private datasets, thereby paving the way for more accurate and personalized diagnoses and treatments while overcoming limitations brought about by a lack of labeled data.

- We propose an Uncertainty and Distance Fusion (UDFusion) strategy to measure the distance between private data and public datasets and thus build reliable pseudo-labels for the student network. The UDFusion separately trains encoder-decoder models on public datasets, which generate various outputs with the same private image to estimate the distribution distances between public datasets and the private image. The UDFusion enhances the teacher networks' adaptability while facilitating the student to efficiently utilize the unlabeled data.

- We propose a Granular Attention Network (GANet) to accurately identify and segment colon polyps. The GANet first identifies polyps roughly from a global perspective by encoding long-range anatomical dependencies. Then, GANet refines the identification and removes false-positive areas by performing multi-scale exploration in every decoding layer.

## Related work

### Colon polyp segmentation

Colon polyp segmentation methods have been proposed in recent years with the development of deep learning. Most of these methods are based on U-Net [5] and its variants [6–8]. For

instance, U-Net is integrated with dilated convolutions to segment colon polyps [9], where the up-down sampling and dilated convolutions enable U-Net to expand the receptive field and extract richer feature information. U-Net++ is employed to extract multi-level and multi-scale features from multiple branches, enabling accurate colon polyp segmentation [10]. ResUNet+ + adds more connections to retain useful features, further improving colon polyp segmentation accuracy [11]. However, due to the diversity of colon polyps across different images, these original U-Net-based methods may produce unsatisfactory results. Thus, some recent methods have been proposed to capture discriminative features to identify diverse polyps effectively. For instance, PolyNet [12] introduces a dual-tree wavelet pooling convolutional neural network to reduce missed high signals. LODNet [13] sets eight derivatives for each pixel and uses the characterization capacity of the derivatives to search for polyps in the boundary region. DCRNet [14] estimates the correlation between image pixels by aggregating contextual information, allowing the model to learn more detailed local discriminative features.

## Unsupervised domain adaptation

Unsupervised domain adaptation (UDA) aims to adapt a model trained on one domain (source domain) to perform well on a new, related domain (target domain) without labeled data for the target domain. UDA has great potential in many medical image segmentation tasks due to the scarcity of labeled data arising from the cost, time, and expertise required for manual annotations. Recent studies on UDA mainly leverage adversarial learning to enforce information alignment. For example, many researchers have developed image alignment methods by adding auxiliary constraints to the CycleGAN framework [15]. SynSeg-Net [16] combined CycleGAN and a segmentation network to form an end-to-end synthetic segmentation method. Zhang et al. [17] proposed a shape-consistency loss to ensure consistent anatomical structures. LE-UDA [18] constructs self-ensembling consistency for knowledge transfer to achieve better feature alignment with CycleGAN. In colon polyp segmentation, some recent UDA methods have also been proposed using the adversarial learning approach, such as DAN-PD [19], or different information alignment approaches [20]. More specifically, Nam et al. [20] first coarsely align the data distributions at the input level using the Fourier transform in chromatic space and then finely align them at the feature level using fine-grained adversarial training.

## Medical segmentation with self-attention

The self-attention mechanism has been widely employed to enable deep neural networks to learn better feature representations and produce more accurate segmentation performance. The self-attention mechanism learns a set of attention scores that dictate how much focus each element in an image should have, allowing for the capture of intricate dependencies in the images [21], thereby contributing to more accurate pixel classification. These intricate dependencies are crucial for identifying structures with complex appearances and textures [22], such as lesions. Therefore, the self-attention mechanism has been widely deployed in medical image segmentation [23]. For instance, the self-attention mechanism is used to segment lung areas in chest X-rays [24], breast cancer in ultrasound images [25], and retinal vessels [26]. For more complex tasks, such as multi-organ segmentation [27, 28], self-attention also significantly improves performance compared to regular convolutional neural networks.

## Teacher-student framework in medical image segmentation

The teacher-student framework has been widely used in medical image segmentation, especially when dealing with unsupervised and semi-supervised learning tasks. Traditionally, the

teacher-student framework is associated with knowledge distillation, where a well-trained teacher model guides a simpler student model. In recent years, the teacher-student framework has been employed in various learning frameworks beyond direct knowledge transfer. For example, Yu et al. [29] utilized the teacher-student framework for semi-supervised medical segmentation, where a teacher network generates pseudo-labels for a student network on unannotated data. Similarly, Chen et al. [30] used the teacher-student framework for semi-supervised COVID-19 infection segmentation. Zhang et al. [31] leveraged the teacher-student framework for non-enhanced tumor segmentation with the assistance of enhanced images, where the teacher network trained with enhanced images generates pseudo-labels for the student network trained with non-enhanced images. In our study, we leverage the core idea of the teacher-student framework—using the output of advanced models to guide the training process of other basic models. We achieve label-free training by generating pseudo-labels from a module trained on three different public datasets (the "teacher") and then using these pseudo-labels to train another independent module (the "student").

## Methods

Our SATS employs a teacher-student framework to integrate the newly proposed UDFusion and GANet, thereby fully utilizing fully annotated public datasets to facilitate colon polyp segmentation on unannotated private data. As illustrated in Fig 2, our SATS builds multiple teacher modules and one student module with the GANet. As indicated by the blue lines, these teacher modules learn to predict the colon polyp mask from the public datasets separately.

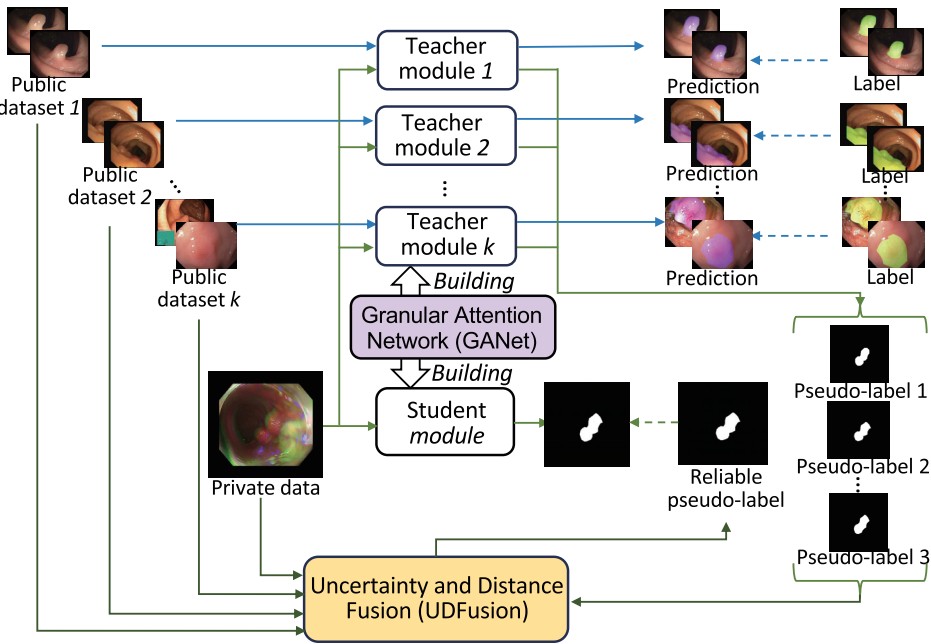

**Fig 2. Our SATS integrated the newly proposed UDFusion and GANet into a teacher-student protocol to achieve the unlabeled colon polyp segmentation by leveraging the labeled public datasets.** Particularly, The GANet is employed to build multiple teacher modules and one student module. The teacher modules are well-trained with public datasets separately, thereby producing multiple pseudo-labels for training the student module on private data. The UDFuion evaluates the distribution between each private image and the public datasets to fuse the multiple pseudo-labels and thus outputs a reliable pseudo-label for the student modules. Therefore, the student module is able to segment private images after training without data annotation.

Simultaneously, as indicated by the green lines, these teacher modules produce multiple pseudo-labels by inputting a private colon polyp image. Then, UDFusion evaluates the distribution distance between the public datasets and private data as well as the uncertainty of the pseudo-labels, thereby fusing the multiple pseudo-labels and generating a reliable pseudo-label. This reliable pseudo-label enables the student module to learn colon polyp segmentation from unannotated private data. Our SATS is validated using three public datasets (CVC-ClinicDB, CVC-ColonDB, and Kvasir) and one private dataset. The private dataset was collected from colonoscopy procedures performed at the First Affiliated Hospital of Anhui Medical University in Hefei, China, from December 30th, 2023, to January 5th, 2024.

## Uncertainty and Distance Fusion (UDFusion)

The UDFusion evaluates the distribution distance between private data and all available datasets and the prediction uncertainty of the teacher modules, thereby outputting a reliable pseudo-label to facilitate the training of the student module. More specifically, assuming $X^i = \{x^i_j\}$ and $Y^i = \{y^i_j\}$ are the images and labels in a public dataset. $i \in [1, k]$ is the index of a dataset and $j \in [1, N_i]$ indicates the dataset contains $N_i$ subjects. $X^{k+1} = \{x^{k+1}_j\}$ and $j \in [1, N_{k+1}]$ indicate the private data and the case number. The reliable pseudo-label by the UDFusion is formulated as:

$$\hat{Y}(p, q) = \sum_{i=1}^{k} w^i(p, q) \times \tag{1}$$

$$= \sum_{i=1}^{k} \frac{u^i + v^i(p, q)}{2} \times \hat{y}^i(p, q) \tag{2}$$

where $\hat{Y}$ is the reliable pseudo-label and $(p, q)$ is the coordinate of each pixel. $w^i$ is the weight of the $i$-th teacher module's prediction $\hat{y}^i$. For each pixel, $\frac{u^i + v^i(p,q)}{2}$ is the average of the distribution weight $u^i$ and the uncertainty weight $v^i(p, q)$. $u^i$ evaluates the reliability of each pseudo label through the contextual information (Eq 5). $v^i(p, q)$ evaluates the reliability of each pseudo label through the pixel information (Eq 6). By integrating the distribution weight and the uncertainty weight, the UDFusion is able to reserve both contextual information of spatial pixels and local information at pixels.

The distribution weight is obtained by inputting a private image into several encoder-decoders and comparing the distance between the outputs and the input image. These encoder-decoders are separately pretrained to fit the distribution of public datasets, where each encoder-decoder restores the input image from one public dataset. Therefore, if we input a private image into these encoder-decoders, the output of an encoder-decoder that is closer to the input private image indicates that the distribution of this private image is closer to the public dataset used to pre-train this encoder-decoder. Assuming each encoder-decoder is denoted as $\phi^i$ where $i \in [1, k]$, the pre-training objective of this encoder-decoder is:

$$\mathcal{J}_{\phi^i} = argmin\, \mathbb{E}[\mathcal{L}_{mse}(\phi^i(x^i_j), x^i_j)] \tag{3}$$

where $\mathcal{L}_{mse}$ is mean square loss. With well-trained encoder-decoders, we can get the distribution distances between a private image $x^{k+1}_j$ and these public datasets:

$$\mathcal{D}(x^{k+1}_j, X^i) = \mathcal{L}_{mse}(\phi^i(x^{k+1}_j), x^{k+1}_j) \tag{4}$$

Therefore, if we shorten $\mathcal{D}(x_j^{k+1}, X^i)$ as $\mathcal{D}^i$, the distribution weight $u^i$ is formulated as:

$$u^i = \text{softmax}(D^i) = \frac{e^{-D^i}}{\sum_{j=1}^{k} e^{-D^i}} \tag{5}$$

Similarly, we can get the uncertainty weight as:

$$v^i(p, q) = \text{softmax}(U^i) = \frac{e^{-U^i(p,q)}}{\sum_{j=1}^{k} e^{-U^j(p,q)}} \tag{6}$$

where $U^i(p, q)$ is the uncertainty of at $(p, q)$ by $\phi^i$. $U^i(p, q)$ is obtain by the information entropy:

$$U^i(p, q) = -\hat{y}^i(p, q) \times \log(\hat{y}^i(p, q)) \tag{7}$$

Entropy is a commonly employed method to effectively measure how scattered or concentrated the model's predictions are, with higher entropy indicating less certainty. More specifically, if $\hat{y}$ is close to 1 or 0, the entropy will be very low, indicating the model can classify the pixel as background or foreground. Otherwise, if $\hat{y}$ is close to 0.5, the entropy will be very high, which indicates the model is uncertain about classifying the pixel as background or foreground.

## Granular Attention Network (GANet)

The GANet encodes the input image to identify colon polyps from a global perspective and then progressively refines the initial prediction from coarse to fine, thus accurately segmenting colon polyps. As shown on the left side of Fig 3, the Identify Transformer Block learns to predict the rough location of polyps, while the Refine Transformer Block gradually makes the

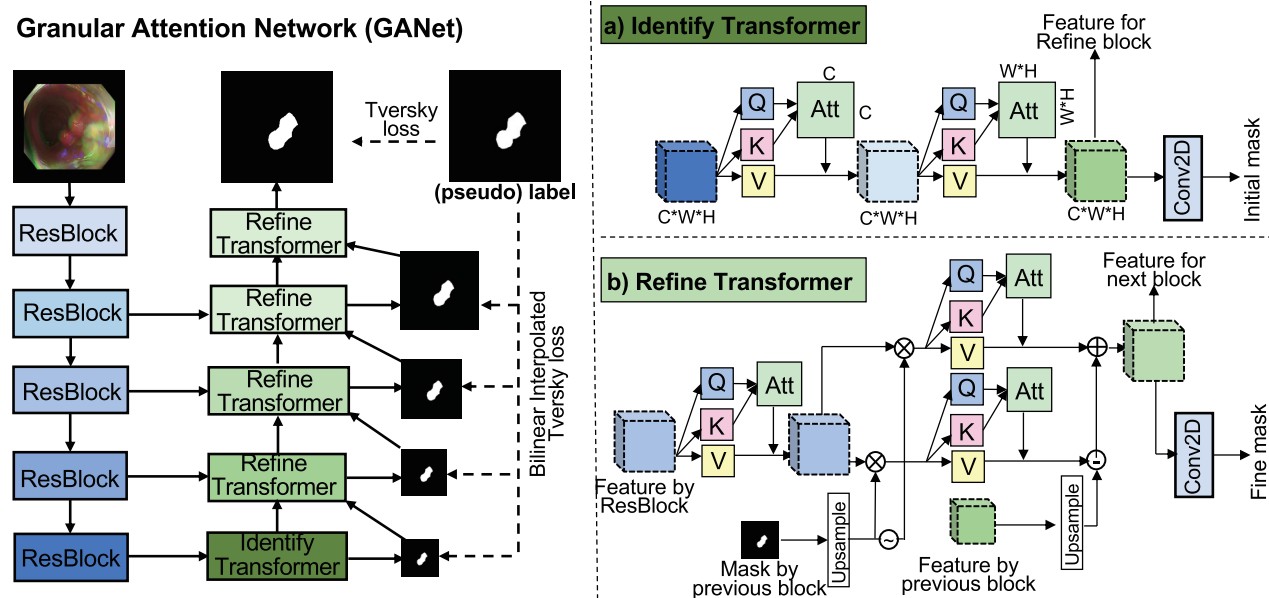

**Fig 3. The architecture of the proposed GANet.** The GANet firstly employs ResBlocks to build the encoder and thus extract features from the input image. Then, the GANet employs the Identify Transformer Block and the Refine Transformer Blocks to build the decoder, thereby segmenting the polyps progressively. Particularly, the Identify Transformer is constructed with a channel-attention module and a spatial-attention module to predict the rough location of the polyps from a global perspective. The Refine Transformer Blocks are constructed with foreground attention and background attention under the guidance of the Identify Transformer Block, thereby predicting more accurate segmentation masks in ambiguous areas.

prediction clearer by integrating the rough prediction from the previous block with the features from ResBlocks. To ensure the Refine Transformer Block focuses on predicting the polyp boundaries instead of the background, GANet employs bilinear interpolated Tversky loss [32] to supervise the training at multiple scales.

**Identify Transformer Block.** As shown in Fig 3a), the Identify Transformer Block collects the highest-level features from the deepest ResBlock, thereby predicting the rough location of polyps and transmitting the prediction with related features to the Refine Transformer Block. The Identify Transformer Block mainly employs the self-attention mechanism to sequentially build a channel-attention module $Attn_{ch}(\cdot)$ and a spatial-attention module $Attn_{sp}(\cdot)$. The channel-attention module enables the Identify Transformer Block to capture the dependence among channels and strengthen the more important channels. For instance, the channel-attention module distributes different weights to the respective channels that represent boundary information and texture information. Assuming that the features from the deepest ResBlock are $z_{D-1} \in \mathbb{R}^{c \times w \times h}$, and $D$ is the number of ResBlocks. The process in the channel-attention module can be summarized as $O_{ch} = Attn_{ch}(z_{D-1})$, which includes:

$$
\begin{cases}
Q_{ch} & = \omega_{ch}^q \otimes z_{D-1} \\
K_{ch} & = \omega_{ch}^k \otimes z_{D-1} \\
V_{ch} & = \omega_{ch}^v \otimes z_{D-1} \\
A_{ch} & = \text{softmax}\left(\dfrac{Q_{ch} \times K_{ch}^T}{\sqrt{c}}\right) \\
O_{ch} & = A_{ch} \times V_{ch}
\end{cases}
$$

Where $Q_{ch}, K_{ch}, V_{ch}$ are the query, key, and value. The query represents the information or relationship a current position seeks from other positions; the key provides an identifier for each position to determine its relevance to the query; the value holds the actual information at each position, which is aggregated once relevance is determined. $\omega$ represents convolutional kernels to be trained, and $\otimes$ indicates convolutional operation. The spatial-attention module enables the Identify Transformer Block to capture the long-term spatial context, allowing the Identify Transformer Block to understand the spatial architecture and shape of desired polyps. The process in the spatial-attention module $O_{sp} = Attn_{sp}(O_{ch})$ is divided into three steps, where the first step is to reshape the input three-dimensional features $O_{ch} \in \mathbb{R}^{c \times w \times h}$ into two-dimensional temporal features $O_{tp} \in \mathbb{R}^{c \times [w \times h]}$:

$$
O_{tp}(i, j \times h + k) = O_{ch}(i, j, k) \tag{8}
$$

where $i \in [0, c-1], j \in [0, w-1]$, and $k \in [0, h-1]$. The second step is to get the attention map and weighted features:

$$
\begin{cases}
Q_{sp} & = \omega_{sp}^q \otimes O_{tp} \\
K_{sp} & = \omega_{sp}^k \otimes O_{tp} \\
V_{sp} & = \omega_{sp}^v \otimes O_{tp} \\
A_{sp} & = \text{softmax}\left(\dfrac{Q_{sp} \times K_{sp}^T}{\sqrt{c}}\right) \\
O_{tp} & = A_{sp} \times V_{sp}
\end{cases}
$$

Same as the channel-attention module, $\omega_{sp}^*$ are the convolutional kernels to be learned. The third step is to reshape the two-dimensional features back into three-dimensional features:

$$O_{sp}(i,j,k) = O_{tp}(i, j \times h + k) \tag{9}$$

where $i \in [0, c-1]$, $j \in [0, w-1]$, and $k \in [0, h-1]$. Then, the feature $O_{sp}$ is sent to the Refine Transformer Block for further finer prediction and is also fed into a convolutional layer to determine the initial location:

$$P_{ini} = \omega_{ini} \otimes O_{sp} \tag{10}$$

**Refine Transformer.** As shown in Fig 3b), the Refine Transformer Block mainly builds foreground attention and background attention with the coarse prediction from the previous block, thereby correcting the falsely predicted area. More specifically, assuming the features from the ResBlock at the same level is $z_d$ and $d \in [0, D-1]$, the up-sampled features from the previous block (Refine Transformer Block or Identify Transformer Block) is $O_{d+1}$, and the up-sampled prediction from the previous block (Refine Transformer Block or Identify Transformer Block) is $P_{d+1}$. Therefore, $O_{D-2}$ and $P_{D-2}$ are from the Identify Transformer Block. Namely, $O_{D-2} = O_{sp}$ and $P_{D-2} = P_{ini}$. The process in the Identify Transformer Block is summarized as follows:

$$\begin{cases} z_d & = Attn_{sp}^{all}(z_d) \\[2mm] z_d^{fg} & = Attn_{sp}^{fg}(P_{d+1} \cdot z_d) \\[2mm] z_d^{bg} & = Attn_{sp}^{bg}((1 - P_{d+1}) \cdot z_d) \\[2mm] z_d & = z_{d+1} - \gamma_1 z_d^{fg} + \gamma_2 z_d^{bg} \\[2mm] P_d & = \omega_d \otimes z_d \end{cases} \tag{11}$$

Where the overall spatial attention $Attn_{sp}^{all}$ is to make the Refine Transformer Block focus on the area that is difficult to classify. The foreground and background attention $Attn_{sp}^{fg}$ and $Attn_{sp}^{bg}$ enable the Refine Transformer Block to capture the spatial dependency of falsely classified foreground and background pixels, thereby correcting them respectively. Particularly, the foreground attention $Attn_{sp}^{fg}$ employs the self-attention mechanism to refine the foreground feature filtered with the coarse prediction by the previous decoder block. Namely,

$$\begin{cases} z_d^{fg} = z_d P_{d+1} \\[2mm] z_d^{fg} = Attn_{sp}^{fg}(z_d^{fg}) \end{cases} \tag{12}$$

where $P_{(}d+1)$ is the polyp mask predicted by the previous decoder block, with intensity 1 as the foreground and intensity 0 as the background; $z_d$ is the feature from ResBlock at the same level. Similarly, we can get the process that the background attention $Attn_{sp}^{bg}$ refines

the background features:

$$
\begin{cases}
z_d^{bg} = z_d(1 - P_{d+1}) \\
z_d^{bg} = \mathrm{Attn}_{sp}^{bg}(z_d^{bg})
\end{cases}
\tag{13}
$$

Because both foreground attention and background attention are based on the self-attention mechanism, here we formulate them as $Attn_sp(f/bg)$ for simplicity. Similar to the spatial-attention module, the process of $Attn_sp(f/bg)$ refining the foreground/background features

$$
\begin{cases}
Q_{f/bg} = \omega_{f/bg}^q \otimes z_d^{f/bg} \\
K_{f/bg} = \omega_{f/bg}^k \otimes z_d^{f/bg} \\
V_{f/bg} = \omega_{f/bg}^v \otimes z_d^{f/bg} \\
A_{f/bg} = \mathrm{softmax}\left(\dfrac{Q_{f/bg} \times K_{f/bg}^T}{\sqrt{c}}\right) \\
z_d^{f/bg} = A_{f/bg} \times V_{f/bg}
\end{cases}
\tag{14}
$$

Where $\omega$ is the convolutional kernel to be learned. Then, $z_{fg}$ is subtracted from $z_{(d+1)}$ because $z_{(d+1)}$ is the coarse foreground feature used by the previous decoder block to predict the previous mask, while $z_{fg}$ is the incorrect foreground feature. $z_{fg}$ is subtracted from $z_{(d+1)}$ to remove the incorrect foreground features from the previous coarse foreground features. The result is then added to $z_{bg}$ because $z_{bg}$ is the incorrect background feature. By adding $z_{bg}$, the foreground feature is completed, thereby predicting a more accurate foreground mask. With these steps, the false foreground prediction is removed, and the false background prediction is retrieved. Finally, a finer prediction is obtained as $P_d$ and delivered to the next Refine Transformer Block along with finer features $z_d$. Our dual-branch strategy, which processes foreground and background features separately, enables each branch to focus more on feature extraction in its respective area. More specifically, the distinction between foreground (polyp) and background (normal tissue) is obvious. By processing them separately, the model can distinguish these regions more effectively, thereby reducing the possibility of misidentification. More importantly, by treating the foreground and background separately, the feature extraction process for each can be tuned more precisely. This allows the model to be more focused and efficient in processing their respective regions, especially when subtle differences in feature expression need to be distinguished.

The loss function in the GANet is a specifically designed multi-scale Tversky loss to constrain the Identify Transformer Block and the Refine Transformer Block to output a more accurate lesion prediction. Particularly, the loss function is formulated as:

$$
\mathcal{L} = \sum_{d=0}^{D-1} \lambda \mathrm{L}_{Tversky}(P_d, \mathrm{I}(y, P_d))
\tag{15}
$$

Where $P_d$ is the lesion prediction from each decoder layer, and $\mathrm{L}_{Tversky}(P_d, y)$ is the Tversky loss. $\mathrm{I}(y, P_d)$ is the bilinear interpolation function that down-samples the label $y$ to the size of $P_d$. $\lambda$ is a coefficient to balance the multi-scale losses. This loss function enables foreground attention and background attention to capture wrong features in both foreground and background regions. Also, the Tversky loss through its parameters provides flexibility to adjust the

weights of false positives and false negatives, which allows us to tune the sensitivity of the segmentation network more finely to the small polyp regions [32].

## Experiment

### Data

The data for validation consists of three public datasets (CVC-ClinicDB, CVC-ColonDB, and Kvasir) and one private dataset. Among the public datasets, CVC-ClinicDB has 612 images collected from 23 cases, CVC-ColonDB has 380 images collected from 13 cases, and Kvasir has 1000 images. The selection of these datasets involves two criteria: the public availability and usage in the community, and the annotation quality. Our private dataset was collected from colonoscopy procedures performed at the First Affiliated Hospital of Anhui Medical University in Hefei, China, from December 30th, 2023, to January 5th, 2024. The data collection was approved by the Institutional Ethics Board. Inclusion criteria consisted of a written agreement to participate and an age of 18 years or older. High-definition endoscopic images were captured, resulting in a dataset comprising 15 cases or 1412 unique images.

For validation and testing, the images of 5 cases (including 485 images) out of the 15 cases were carefully reviewed by a team of expert gastroenterologists to annotate the polyp boundaries. All patient information was anonymized to protect patient privacy. Each public dataset is divided into 90% for training and 10% for validation, thereby obtaining the corresponding optimal teacher module. In the private data, the images of 10 unannotated cases (including 927 images) were used for training, the images of 2 annotated cases (including 183 images) were used for validation, and the images of the remaining 3 annotated cases (including 202 images) were employed for independent testing.

Note that we did not employ any pre-processing methods to augment the public datasets. The core innovation of our method is to leverage the distribution biases among different datasets to generate pseudo-labels of varying quality. Instead of pre-processing to enhance the pseudo-label quality, we embrace these differences as they provide a rich source of information that our framework can utilize.

### Implementation

The implementation details are divided into three parts: Environment configuration, hyper-parameter settings, and evaluation matrices. Environment configuration: The framework is implemented with Pytorch-1.6 (https://pytorch.org/) with CUDA-10.2 and cuDNN-7.6.5 on 4 NVIDIA Tesla V100 PCIe 32-GB GPUs under Ubuntu 20.04.

Hyper-parameter settings: The total training epoch number is 200. Adam optimizer is employed with 2e-4 as the initial learning rate and 0.9 and 0.999 as the decay rates for the first and second gradient moments. The batch size is set as 4. The weight coefficient $\lambda$ in the multi-scale Tversky loss is set as $2^d$, where $d$ is the index of decoders, including the Location and Refine Transformer Blocks. More selection details about the learning rate and weight coefficient can be found in the experimental results.

Performance evaluation matrices: Seven matrices, including Intersection-over-Union (IoU), Dice coefficient (Dice), Precision, Recall, Accuracy, Hausdorff Distance (HD), and Mean Surface Distance (MSD), are employed to evaluate the method's performance.

### Experimental results

**Qualitative evaluation.**  As shown in Fig 4, although the colon polyps present various shapes and appearances, our method segments colon polyps accurately with a high consistency

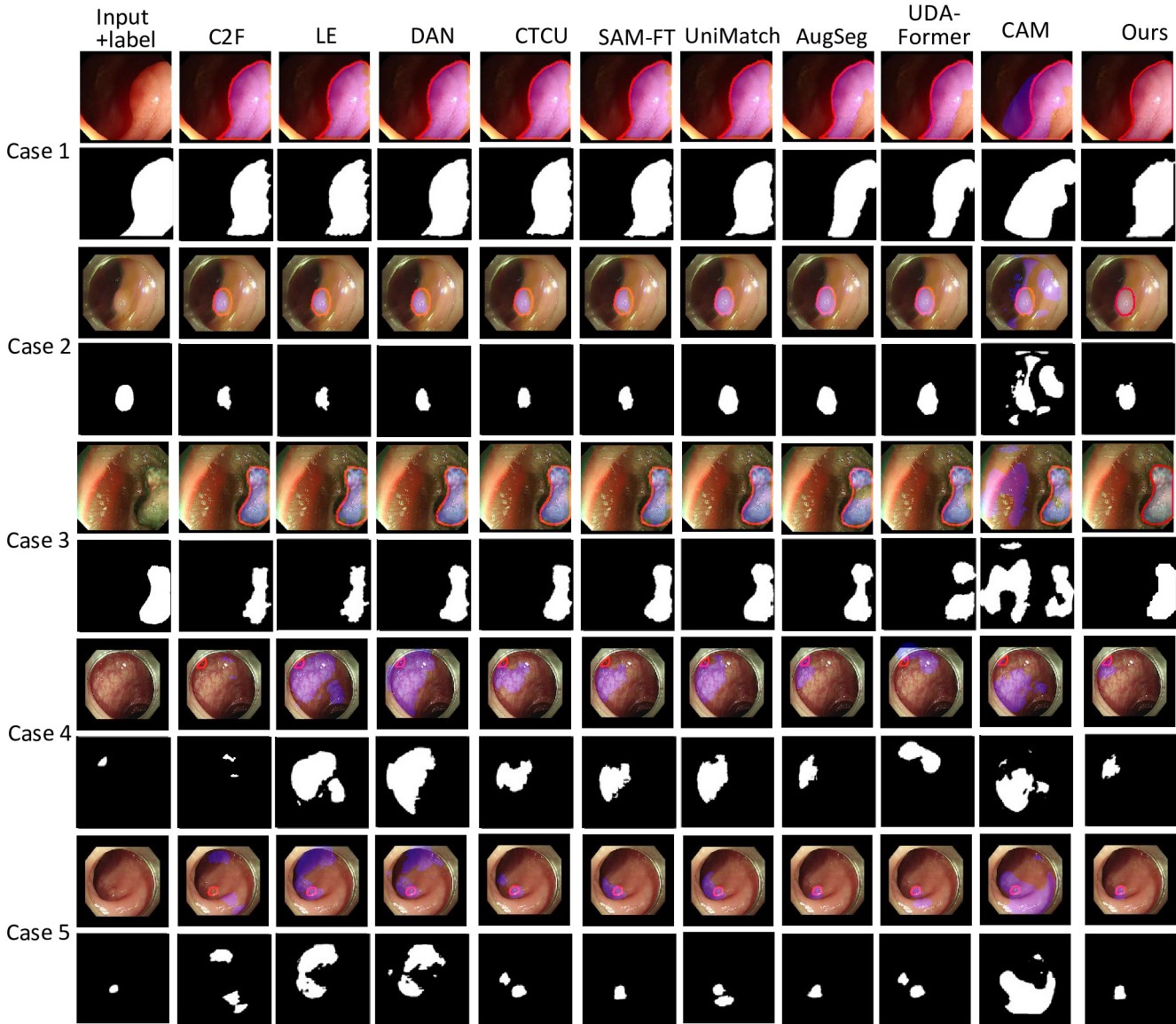

**Fig 4. Five visualized examples to demonstrate the superiority of our SATS compared with existing colon polyp segmentation methods.** The color images are the polyp images, and the binary images are the labels or segmentation results. In each polyp image, the red boundary is the label, and the blue mask is the segmentation.

between the predicted mask and the label. For instance, the polyp in case 3 presents a different appearance and texture compared with other cases, but our SATS captures the difference between the polyp and the background, segmenting the polyp with a high accuracy. The polyp in case 4 is totally blurred with surrounding tissues, but our SATS accurately identifies the polyp and accurately predicts the polyp boundary.

**Quantitative evaluation.** As illustrated in Table 1, our SATS overall achieves a great performance in terms of the seven evaluation matrices. More specifically, the SATS obtains 76.30 ±13.06 on IoU and 85.67±9.76 on Dice, indicating the prediction by our SATS is highly overlapped with the ground truth. The SATS achieves 3.48±0.49 on MSD, indicating the boundary by the SATS is highly consistent with manual delineation. These results demonstrate that our

**Table 1. Our SATS attains a new state-of-the-art performance in the unsupervised colon polyp segmentation than all five comparative methods.**

|  | IoU ↑ | Dice ↑ | Precision ↑ | Recall ↑ | Accuracy ↑ | HD ↓ | MSD ↓ |
|---|---|---|---|---|---|---|---|
| C2F | 70.14 | 76.54 | 87.58 | 70.51 | 97.21 | 13.53 | 6.06 |
| [20] | ±16.50 | ±17.03 | ±18.46 | ±16.52 | ±0.86 | ±2.88 | ±1.37 |
| LE | 67.18 | 74.93 | 89.82 | 67.16 | 98.27 | 15.21 | 7.83 |
| [18] | ±30.68 | ±30.32 | ±20.25 | ±30.66 | ±0.02 | ±17.92 | ±12.42 |
| DAN | 69.87 | 76.06 | 86.84 | 69.88 | 98.23 | 13.79 | 6.25 |
| [19] | ±32.86 | ±32.71 | ±33.86 | ±32.87 | ±0.02 | ±17.93 | ± 15.62 |
| CTCU | 74.18 | 80.78 | 91.32 | 74.36 | 98.33 | 12.29 | 4.68 |
| [33] | ±17.48 | ±17.23 | ±18.15 | ±18.48 | ±0.01 | ±4.12 | ±1.75 |
| SAM-FT | 70.87 | 76.52 | 86.79 | 70.90 | **98.80** | 10.53 | 5.12 |
| [34] | ±23.62 | ±23.30 | ±23.85 | ±23.92 | **±0.01** | ±12.17 | ±8.58 |
| UniMatch | 71.85 | 83.7 | 90.25 | 71.86 | 98.61 | 12.67 | 5.40 |
| [36] | ±24.33 | ±24.21 | ±23.78 | ±24.91 | ±0.06 | ±8.39 | ±5.34 |
| AugSeg | 73.29 | 84.55 | **93.37** | 73.30 | 98.68 | 9.65 | 4.45 |
| [37] | ±15.22 | ±10.57 | **±6.10** | ±13.25 | ±0.02 | ±2.78 | ±1.21 |
| UDA-Former | 72.79 | 84.25 | 91.98 | 72.81 | 98.54 | 10.15 | 5.23 |
| [38] | ±17.75 | ±18.31 | ±20.49 | ±12.27 | ±0.08 | ±6.48 | ±2.47 |
| CAM | 52.72 | 69.04 | 63.89 | 52.72 | 97.09 | 15.49 | 8.63 |
| [35] | ±40.48 | ±37.21 | ±34.81 | ±35.53 | ±0.57 | ±12.17 | ±16.44 |
| **SATS** | **76.30** | **85.67** | 89.98 | **86.00** | 98.36 | **7.01** | **3.48** |
| **(Ours)** | **±13.06** | **±9.76** | ±12.92 | **±16.75** | ±0.12 | **±1.09** | **±0.49** |

SATS is effective in unlabelled colon polyp segmentation by utilizing the annotated public datasets.

**Comparison experiments.** As demonstrated in Fig 4 and Table 1, our SATS outperforms five state-of-the-art methods in this unlabelled colon polyp segmentation task. Particularly, among the nine existing methods, C2F [20], LE [18], DAN [19], and CTCU [33] are based on unsupervised domain-adaptation. The UniMatch, AugSeg, and UDA-Former are three semi-supervised methods. The above seven methods attempt to transfer the knowledge learned from fully-labelled colon polyp datasets to unlabelled datasets. SAM-FT [34] is fine-tuned using fully labeled colon polyp datasets and subsequently tested on our private, unlabeled data. The CAM [35] leverages image-level labels (with polyp or without polyp) to achieve pixel-level segmentation. All these methods are trained with the same dataset as ours to ensure fairness.

As the visualized result demonstrates in Fig 4, our method is more competitive than the five existing methods in this unlabeled colon polyp segmentation task. For instance, in case 3, the nine existing methods fail to distinguish the polyp boundary due to the complex light conditions and polyp textures. In case 5, some existing methods are unable to find the correct area of the polyp due to the unique perspective of the endoscope. In comparison, our SATS obtains a more accurate result, where the predicted boundary is smooth and highly consistent with the manual label.

As the quantitative result illustrates in Table 1, our SATS outperforms the nine existing methods in terms of five evaluation matrices out of the seven matrices. More specifically, the CAM obtains the poorest with 52.72% on IoU and Recall. Such a poor result may be because the LE attempts to transform the styles of source domains (public data) and target domains (private data), but the data size is too small to facilitate it. Although CAM achieves a better result with 52.72% of Dice, it is unsatisfactory due to it only involves image-level information.

The other seven existing methods, including the C2F, DAN, and SAM, achieve similar performance with about 70% on IoU and Precision, which are lower than our SATS by about 6% on IoU and 16% on Recall, respectively. The above comparison indicates our SATS is the best one to segment colon polyps by leveraging annotated datasets.

**Ablation studies.** The ablation studies evaluate the effectiveness of our proposed UDFusion and GANet separately. More specifically, the ablation studies first combine the UDFusion with various segmentation networks to demonstrate the superiority of our GANet. The segmentation networks include Mob-PNet, DSNet, UNet, and ResUNet.

As shown in Figs 5 and 6, our GANet is more effective than existing colon polyp segmentation networks in this task. More specifically, in Fig 5(a), the mean Dice value by GANet is much larger than existing colon polyp segmentation networks; the standard deviation of Dice is much shorter than the existing networks. The standard deviation values by Mob-PNet and UNet are even larger than the corresponding mean values, indicating these two networks are so unstable that they produce several totally false segmentation results. The ResUNet and

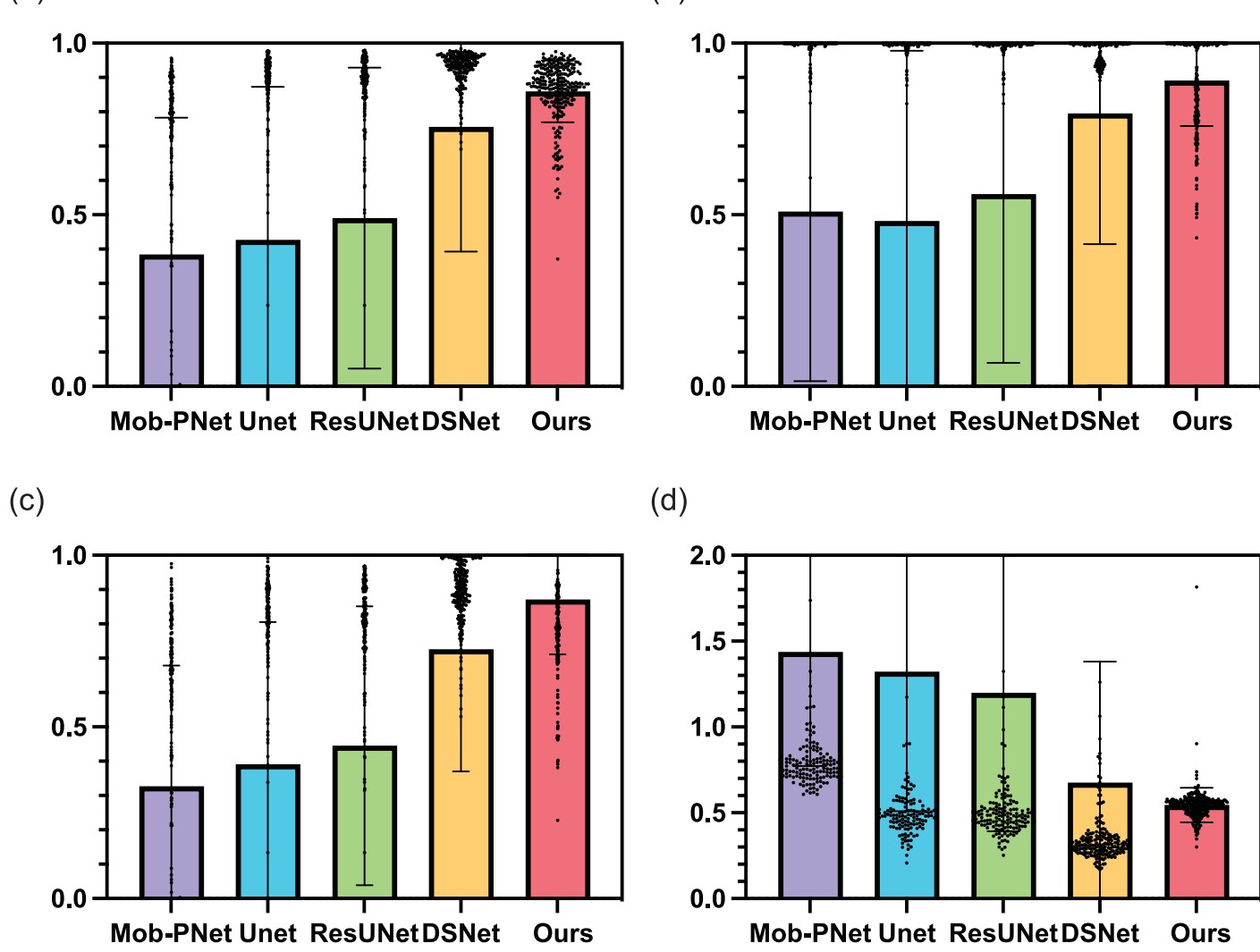

**Fig 5. The ablation study combines UDFusion with various colon polyp segmentation networks to demonstrate the superiority of our GANet.**

DSNet obtain about 0.5 and 0.75 in Dice, respectively, which is much lower than our GANet. In Fig 5(b) and 5(c), our GANet produces higher Precision and Recall values compared with the other four networks. The Mob-PNet, UNet, and ResUNet present similar distributions as their Dice values, where the mean values are much smaller than the value by our GANet, and the standard deviation values are much larger than the value by our GANet. Although DSNet obtains a better result compared with the above three networks, its standard value is much higher than the value of our GANet, indicating it is less stable than our GANet. In Fig 5(d), due to the Mob-PNet, UNet, and ResUNet producing some false predictions, the range of their MSD values is much larger than our GANet. The mean MSD value by the DSNet is larger than the value by our GANet, demonstrating our GANet produces more accurate boundaries than the DSNet.

As the qualitative comparison shown in Fig 6, our GANet is more capable than the other four networks in the polyp identification and segmentation. More specifically, in case 1, the existing four segmentation networks fail to identify the polyp from surrounding tissues, while our GANet is the only network predicting the polyp location correctly. In case 2 and case 3, our GANet generates a polyp mask that is more consistent with the ground truth than the other four networks. All these results demonstrate that our GANet is superior to existing networks in this colon polyp segmentation task.

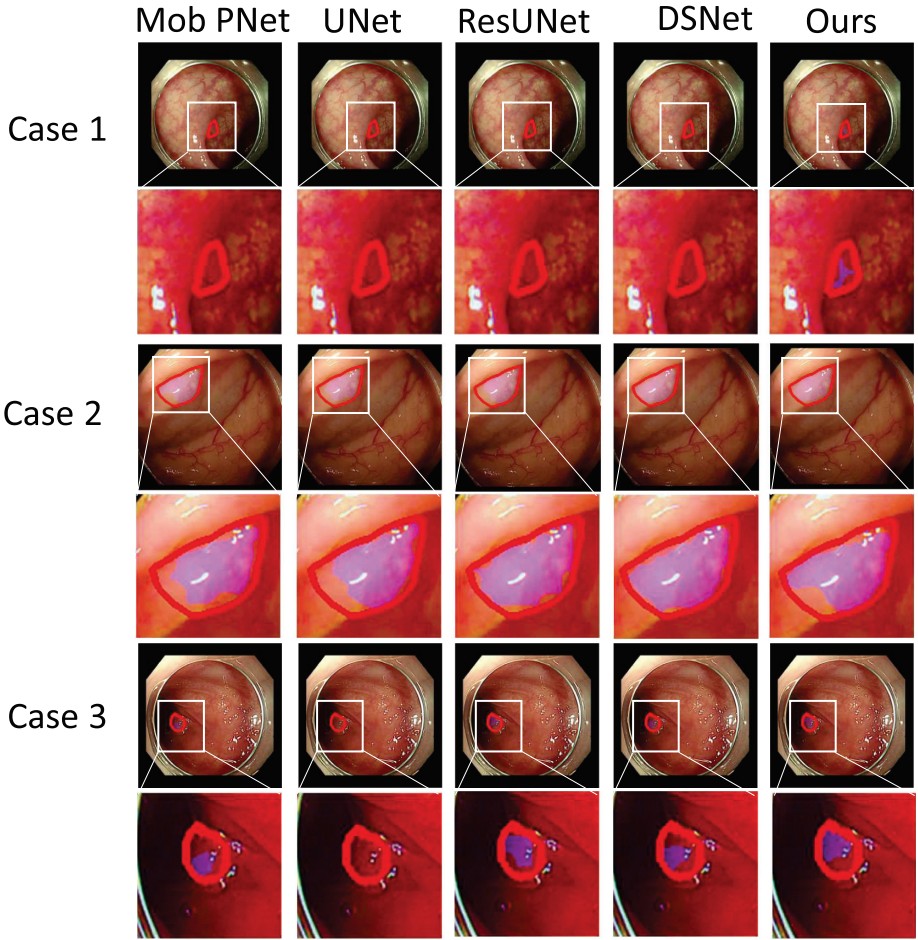

**Fig 6. Three examples to demonstrate that our GANet is more capable of colon polyp segmentation compared with the existing four networks.**

**Table 2. Ablation studies to validate the necessity of the Identify Transformer Block and the Refine Transformer Block.**

| Method | Dice | Prec | Acc | MSD |
|---|---|---|---|---|
| Baseline (ResUNet) | 49.18 | 56.20 | 95.26 | 57.30 |
| Baseline w/ Identify Transformer | 75.70 | 79.58 | 98.68 | 6.03 |
| Baseline w/ Refine Transformer | 78.88 | 79.26 | 98.87 | 7.29 |
| Identify & Refine Transformers (Ours) | 86.67 | 89.98 | 98.36 | 3.48 |

As shown in Table 2, we employ the ResUNet as the baseline and validate the improvement by the Identify Transformer Block and the Refine Transformer Block, thereby showing the necessity of these two components. Particularly, compared with the baseline method, the Identify Transformer Block and the Refine Transformer Block significantly improve segmentation performance in terms of the four evaluation matrices. For instance, the Identify Transformer Block enables a Dice of 75.70%, which outperforms the baseline result by over 25%. Similarly, the Refine Transformer Block enables a Dice of 78.88%, outperforming the baseline result by over 28%. Our method obtains the best performance by integrating the Identify Transformer Block and the Refine Transformer Block.

The ablation studies also combine GANet with different pseudo-label generation methods to demonstrate the advantage of our UDFusion. These different pseudo-label generation methods include 1) Merging all public datasets together to train a unified model and taking its predictions on the private data as pseudo-labels (i.e., Merge); 2) Separately training models on public datasets and averaging their predictions as pseudo-labels (i.e., Mean); 3) Only keeping the uncertainty part in our UDFusion (i.e., Unctn); 4) Only keep the distance part in our UDFusion (i.e., Dist). As shown in Figs 7 and 8, our UDFusion outperforms these pseudo-label generation methods quantitatively and qualitatively. Particularly, as Fig 7(a) demonstrates, the Merge obtains the lowest Dice with a value of about 0.7. The Mean, Unctn, and Dist obtain similar results with the mean values less than 0.8. However, their standard deviation values are about 0.3, indicating these methods are unstable. Our UDFusion combining both uncertainty and distance evaluation, achieves the largest mean Dice value and the smallest standard deviation value. Similarly, for Precision and Recall in Fig 7(b) and 7(c), our UDFusion still obtains higher mean values and lower standard deviation values compared with the other four methods. For instance, the Unctn and the Dist obtain mean Precision values of 0.8 and mean Recall values of 0.75. Our UDFusion, as the combination of these two methods, effectively integrates their advantages and achieves mean values of about 0.9 on Precision and Recall.

As the visualized examples demonstrate in Fig 8, qualitatively, our UDFusion predicts a mask more consistent with the ground truth compared with the other four methods. For instance, in case 2 and case 3, the predictions by the four methods are largely different from the ground truth due to the low contrast between surrounding tissues and the polyp. Namely, these four methods present a quite high false negative rate inside the polyp area. However, in comparison, the prediction by our UDFusion is close to the contour by ground truth.

**Optimal hyper-parameter selection.** The optimal hyper-parameter selection focuses on the determination of the coefficient in the multi-scale Tversky loss and the learning rate of GANet. More specifically, for the coefficient in the multi-scale Tversky loss, a bigger prediction expects a larger coefficient since it is closer to the ultimate prediction. Thus, the selection experiment sets the coefficient as $[1, d, 2^d, 3^d]$ respectively and observes the performance. As shown in Fig 9, as the coefficient increases from 1 to $2^d$, the Dice value is improved gradually,

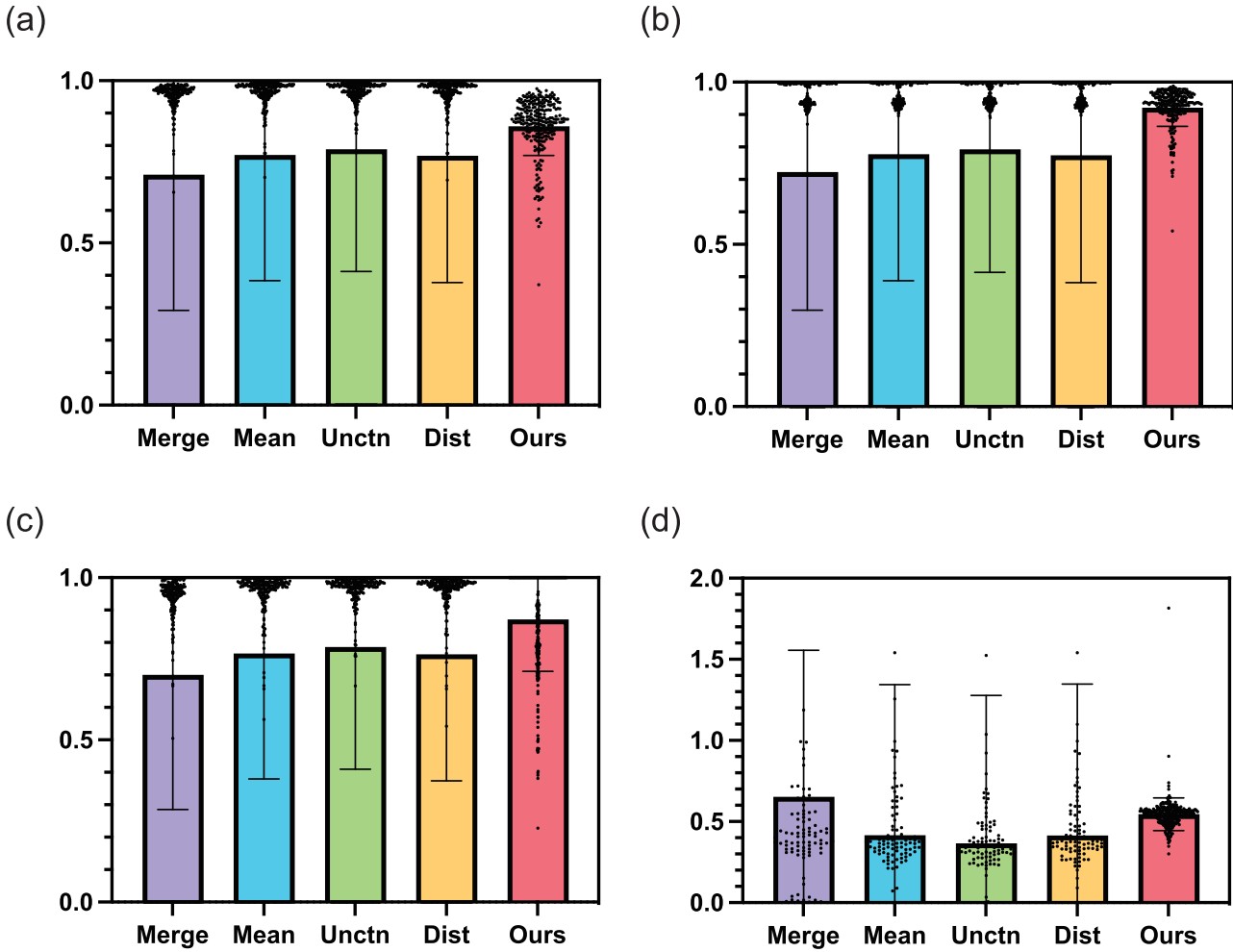

**Fig 7. The ablation study combines our GANet with various pseudo-label generation methods to demonstrate the superiority of our UDFusion.**

and the MSD value is decreased. However, if the coefficient for the larger prediction is too large, i.e., $3^d$, the segmentation performance decreases. This is because the smaller predictions are distributed with too small coefficients, which causes the smaller predictions to be ineffective in facilitating the ultimate prediction performance.

The experiment also compares the segmentation performance with various learning rates between $1e-3$ and $1e-4$, which indicates that $2e-4$ is the optimal learning rate. More specifically, as demonstrated by Fig 10 when the learning rate is set as $1e-3$, the segmentation performance is poor with a Dice value of 0.4, indicating such a large learning rate causes the unstable of network training, especially for the transformer under a pseudo-label task. As the learning rate decreases, the segmentation performance is improved. When the learning rate is set as $2e-4$, the segmentation performance achieves the best with the highest Dice value and the lowest MSD value. However, when the learning rate is smaller, the performance decreases to some extent. The reason for such a decrease is the network is easily stuck into local optimal with a too-small learning rate.

We also employ fewer teacher networks and fewer public datasets to observe the effectiveness of the teacher network numbers. As shown in Table 3, when there is one teacher network

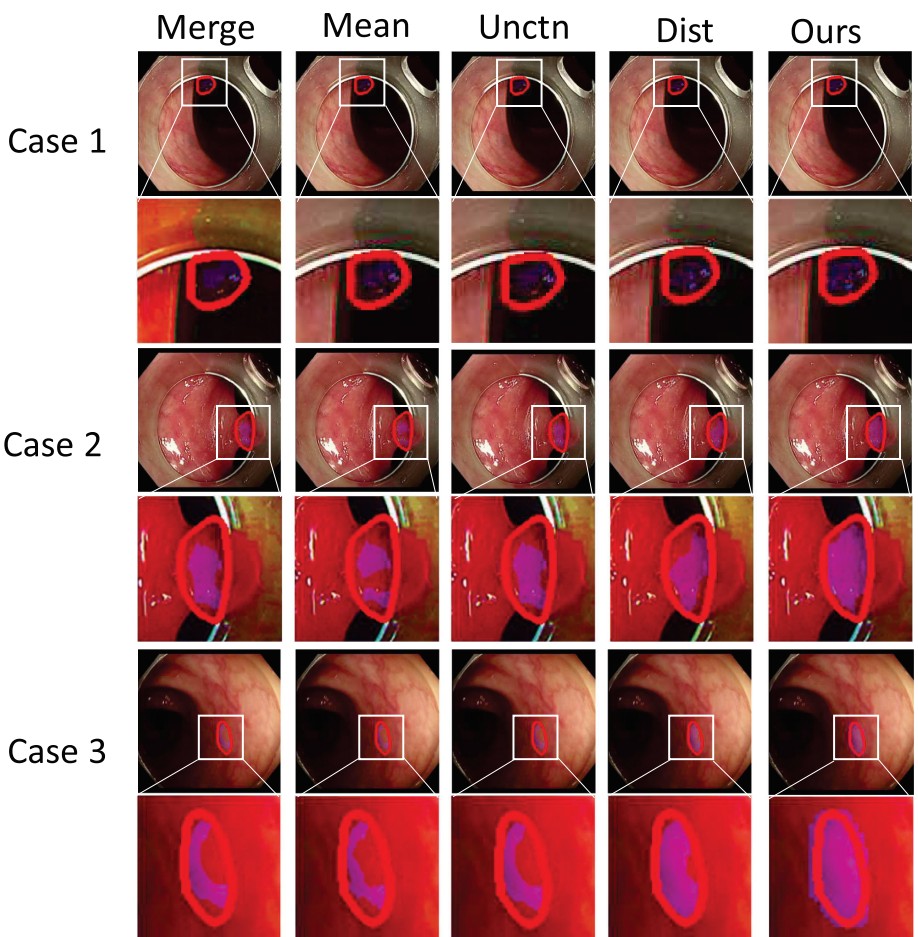

**Fig 8. Three visualized segmentation examples to demonstrate our UDFusion outperforming the other four pseudo-label generation methods.**

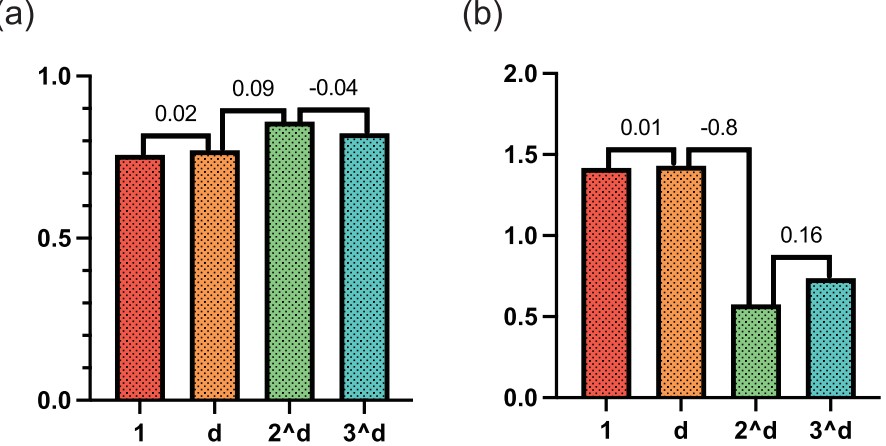

**Fig 9. The experiment indicates $2^d$ is the optimal for the coefficient in the multi-scale Tversky loss.**

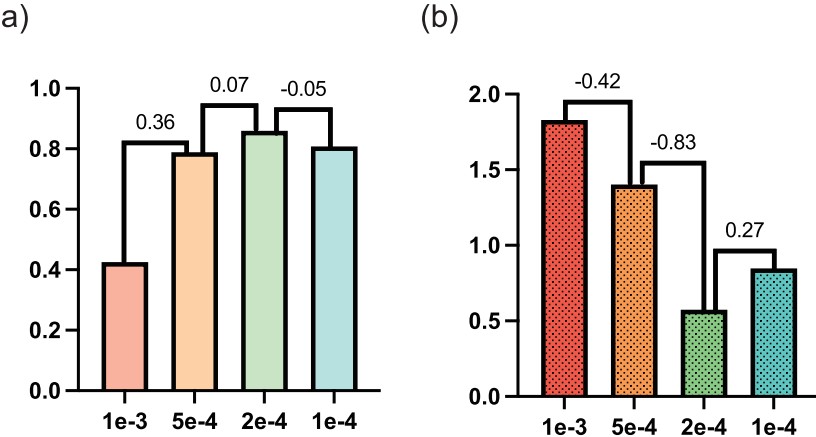

**Fig 10. The experiment indicates $2e-4$ is the optimal for the learning rate.**

or one public dataset available, the Kvasir dataset facilitates the best performance compared with the other two datasets, indicating that the distribution of the Kvasir dataset is closet to our private data. With more teacher networks or public datasets being included, the performance is increasing. For instance, when there are CVC-ColonDB and Kvasir available, the segmentation performance is 85.71% of Dice, 75.51% of Recall, and 5.72 pixels of MSD. With all three public datasets, our method achieves the best performance, which is 86.67% of Dice, and 3.48 pixels of MSD, indicating the importance of teacher network numbers.

## Discussion

Our SATS is more effective than other methods in addressing the distribution-difference issue of colon polyp segmentation. As the comparison and ablation study results indicate, existing methods exhibit a large standard deviation on the evaluation metrics, and ablation methods even fail to identify the colon polyps, illustrating the significant distribution differences between private and public datasets. Compared with these methods, our SATS achieves smaller standard deviations in each evaluation metric and correctly identifies each colon polyp. These results illustrate the superiority of our SATS in fully exploiting the various distributions among public datasets and facilitating the complex distribution of private data. Our SATS can be further integrated with federated learning, thereby applying the proposed protocol to other applications without public datasets. Specifically, during the training of the teacher modules, the SATS deploys each teacher module on each available dataset without involving other

**Table 3. Segmentation results with fewer datasets and teacher networks.**

| Datasets | | | Evaluation | | |
|---|---|---|---|---|---|
| ClinicDB | ColonDB | Kvasir | Dice | Recall | MSD |
| ✓ | | | 78.46 | 64.56 | 8.71 |
| | ✓ | | 78.19 | 64.19 | 7.91 |
| | | ✓ | 80.89 | 67.91 | 7.36 |
| ✓ | ✓ | | 79.74 | 66.30 | 7.59 |
| ✓ | | ✓ | 83.18 | 71.21 | 5.83 |
| | ✓ | ✓ | 85.71 | 75.01 | 5.72 |
| ✓ | ✓ | ✓ | 86.67 | 86.00 | 3.48 |

datasets. During the training of the student modules, the SATS only employs the well-trained teacher modules. In this case, there is no direct interaction between the public datasets and the private dataset, indicating our protocol meets the basic rules of federated learning. Thus, our protocol can be extended under a federated learning framework.

Although our SATS achieves satisfactory results in the colon polyp segmentation task, it has a few limitations. 1) Two-Step Method: Our SATS is a two-step method, which takes more training effort and time than end-to-end methods. Specifically, the UDFusion in the SATS pre-trains reconstruction-based encoder-decoders to measure the prediction uncertainty of teacher modules and the distribution similarity between public datasets and private data. This pre-train step requires manual model selection. 2) Computational Resources: Our SATS deploys the self-attention mechanism on large-scale features, which requires significant computing resources. Specifically, the GANet employs two self-attention mechanisms in each decoder layer. Compared with regular methods that only employ self-attention mechanisms in the deepest decoder layer, our GANet requires more GPU resources, which may limit the application range of our method. 3) Limited Public Datasets: Our method currently involves only three public datasets, resulting in varied prediction results from multiple teacher networks on private data. This can cause the fused pseudo-labels to be unclear and appear mosaic-like, leading to unsatisfactory results in some cases. More specifically, as illustrated by Eq 1, the fewer public datasets lead to less comprehensive pseudo-labels, which further cause unstable prediction in the validation stage.

In our future work, we will focus on two aspects. Data Collection: We will collect more data to improve segmentation performance, including a wider variety of polyp images, enabling the segmentation model to learn a broader distribution and thus enhance segmentation performance. Simplified Network Architecture: We will simplify the network architecture to achieve lower complexity. With a large amount of data, we will modify the architecture using more advanced tools, such as the vision transformer, thereby reducing the complexity of the network. A simplified architecture will also facilitate the deployment of our methods in clinical validation.

## Conclusion

In this paper, we propose SATS to segment colon polyps from unannotated private data by making full use of multiple public polyp datasets. The SATS consists of the newly proposed UDFusion to build more reliable pseudo labels, which adaptively determines the weights of the multiple predicted masks with a novel self-supervised similarity evaluation strategy. The SATS also consists of the novel GANet to predict more accurate segmentation results, which is designed to simulate clinicians' process that identifies polyps first and then focuses on ambiguous areas to refine the coarse prediction progressively. The SATS is validated with three public datasets and one private dataset. Our method achieves 76.30% on IoU, 86.00% on Recall, and 7.01 on HD, which outperforms the existing five methods, indicating the effectiveness of this method on colon polyp segmentation.

## Author Contributions

**Conceptualization:** Yiwen Jia, Tang Yang.

**Data curation:** Guangming Feng, Tang Yang, Siyuan Chen, Fu Dai.

**Funding acquisition:** Fu Dai.

**Investigation:** Siyuan Chen.

**Methodology:** Yiwen Jia, Siyuan Chen.

**Resources:** Guangming Feng, Fu Dai.

**Software:** Yiwen Jia, Tang Yang.

**Supervision:** Fu Dai.

**Validation:** Yiwen Jia, Guangming Feng, Siyuan Chen.

**Visualization:** Tang Yang.

**Writing – original draft:** Yiwen Jia, Guangming Feng.

**Writing – review & editing:** Yiwen Jia, Guangming Feng, Tang Yang, Siyuan Chen, Fu Dai.

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
