## [Decision Letter · Decision Letter 0]

3 Apr 2024

PONE-D-24-03994Self-Adaptive Teacher-Student Framework for Colon Polyp Segmentation from Unannotated Private Data with Public Annotated DatasetsPLOS ONE

Dear Dr. Dai,

Thank you for submitting your manuscript to PLOS ONE. After careful consideration, we feel that it has merit but does not fully meet PLOS ONE’s publication criteria as it currently stands. Therefore, we invite you to submit a revised version of the manuscript that addresses the points raised during the review process.

**ACADEMIC EDITOR: ** Does the information on segmentation in areas other than the colon polyp not enter into this consideration?

We look forward to receiving your revised manuscript.

Kind regards,

Kazunori Nagasaka

Academic Editor

PLOS ONE

Journal Requirements:

"This work is supported by the research funding of Anhui Medical University (2022xkj105)."

"This work is supported by the research funding of Anhui Medical University (2022xkj105)."

"This work is supported by the research funding of Anhui Medical University (2022xkj105)."

5. We note that you have indicated that there are restrictions to data sharing for this study. For studies involving human research participant data or other sensitive data, we encourage authors to share de-identified or anonymized data. However, when data cannot be publicly shared for ethical reasons, we allow authors to make their data sets available upon request. For information on unacceptable data access restrictions, please see http://journals.plos.org/plosone/s/data-availability#loc-unacceptable-data-access-restrictions. 

6. Please ensure that you refer to Figure 10 in your text as, if accepted, production will need this reference to link the reader to the figure.

**Additional Editor Comments:**

Dear Authors,

Thank you very much for submitting your research to PLOS ONE.

Ou decision is major revision.

Please reply to all reviewer's comments and revise the manuscript accordingly.

Sincerely,

Kazunori Nagasaka

Reviewers' comments:

Reviewer's Responses to Questions

**Comments to the Author**

1. Is the manuscript technically sound, and do the data support the conclusions?

Reviewer #1: Yes

Reviewer #2: Yes

Reviewer #3: Partly

2. Has the statistical analysis been performed appropriately and rigorously? 

Reviewer #1: Yes

Reviewer #2: Yes

Reviewer #3: Yes

3. Have the authors made all data underlying the findings in their manuscript fully available?

Reviewer #1: No

Reviewer #2: No

Reviewer #3: Yes

4. Is the manuscript presented in an intelligible fashion and written in standard English?

Reviewer #1: Yes

Reviewer #2: Yes

Reviewer #3: Yes

5. Review Comments to the Author

Reviewer #1: This paper proposed a self-adaptive teacher-student framework for polyp segmentation, to bridge the gap between the annotated datasets and the unannotated datasets. The paper is well-structured but still need further improvement.

1. The major problem of this paper is the experiment on the private dataset. Among the public datasets, CVC-ClinicDB has 612 cases, CVC-ColonDB has 380 cases and Kvasir has 998 cases. However, the private dataset only has 15 cases and only 3 cases were used for model testing. Since this paper aims to solve the data distribution gap between the well-annotated public datasets and the unannotated private dataset. I am afraid such a small-scale cannot cover the heterogeneity of the polyps. How to select the sample to avoid sample selection bias? Authors should carefully response this comment, otherwise I may not be able to accept this paper due to the unconvincing experimental results.

2. In Fig. 6, the results from different methods seems have no different. In case 2, I also wonder whether authors show the same frame? Because the image content of the highlighted areas in different approaches are obvious different. But the segmentation results seem all the same. I guess the red boundaries might be the GT, then where are the segmentation results? Can authors explain this?

2. Why two proposed modules UDFusion and GANet are not introduced in abstract?

3. The captions of figures should deliver more information to help readers better understand the technical novelties, especially Fig. 2 and Fig. 3

4. Besides SAM, authors only compared their proposed model with four competitors proposed in 2022. I suggest comparing more recent approaches proposed in 2023.

5. English should be improved.

For example in abstract: “builds pseudo labels” should be “generates pseudo labels”

“innovatively bridging the gap” “innovatively” should be removed.

“This strategy mainly dynamically” “mainly” should be removed.

Reviewer #2: This paper introduces the Self-Adaptive Teacher-Student (SATS) framework, an unsupervised approach for colon polyp segmentation based on a teacher-student architecture. The SATS method comprises two primary phases. In the initial phase, pseudo-labels are generated by training three teacher networks on three existing public datasets. Using unannotated private data as input, the Uncertainty and Distance Fusion (UDFusion) method processes the outputs of teacher networks to obtain adaptive pseudo-labels. The second phase involves training a student network. Pseudo-labels generated in the prior phase are utilized to train a Grainy Attention Network (GANet), ensuring accurate identification and segmentation of colon polyps. Despite achieving satisfactory results, the paper still faces challenges in terms of method description and experimental aspects:

Comment #1:

Is the term "Teacher-Student" framework appropriate for this paper? Because this study involves a two-step unsupervised semantic segmentation process. Initially, three networks are trained using three public datasets, and pseudo-labels are generated using these trained networks. Subsequently, a separate network is trained using these pseudo-labels. The so-called teacher-student networks do not participate in the same framework for training and prediction. Moreover, they share identical structures.

Comment #2:

It appears that Figure 1(b) lacks a specific explanation of the feature distribution plot. Could you please clarify the meanings of the horizontal and vertical axes?

Comment #3:

Is the notation 'wivi(p, q)' on line 158 written incorrectly and should it be 'wi(p, q)'?

On line 173, is the description 'of at (p, q)' correct?

Comment #4:

The Uncertainty and Distance Fusion (UDFusion) method calculates weights on a per-pixel basis, lacking consideration for the correlation between pixels. Does this lead to the pseudo-label images losing contextual information of spatial pixels? It is hoped that the authors can provide an explanation for this issue.

Comment #5:

Formulas 6 and 7 provide limited descriptions of the uncertainty weight. Can there be an appropriate addition to the explanation of the interpretability of the uncertainty weight and the reasons behind its design?

Comment #6:

The correspondence between the modules in the architecture of Figure 3 and the described content in the text is not clearly illustrated. Additionally, the explanation in the text regarding the 'foreground attention' and 'background attention' of the Refine Transformer Block, and how they continuously correct mispredicted areas, is not sufficiently thorough.

Comment #7:

In Formula 11, why is 'zd' subtracted from 'zfg' and added to 'zbg'? Both 'zfg' and 'zbg' are calculated from the same two feature maps, with the only difference being the sign choice for Pd+1. What is the significance of the dual branches?

Comment #8:

In Figure 3, there is an input 'Od+1' in the 'zfg' branch, but there is no explanation for the 'Od+1' input from Figure 3 in Formula 11.

Comment #9:

Most of the visualization examples in Figure 4 are cases where the polyp shapes have smooth and regular boundaries. How does the SATS method perform in cases with complex boundaries? In Figure 4, for cases 1 and 3, the segmentation edges obtained by the method in this paper are not clear and appear mosaic-like. What could be the reason for this?

Comment #10:

In Table 1, were the other comparison experiments conducted using the same networks as those proposed in this paper, and were they also trained using the three public datasets? How was fairness ensured in the comparison? How do the parameter count and complexity of the compared methods compare? It is recommended to include some networks with similar attention structures in the comparison experiments for more convincing results.

For example:

1)Yu C, Xiao B, Gao C, et al. Lite-hrnet: A lightweight high-resolution network[C]//Proceedings of the IEEE/CVF conference on computer vision and pattern recognition. 2021: 10440-10450.

2)Haiqiang Liu, Meibao Yao*, Xueming Xiao, Bo Zheng, Hutao Cui. MarsScapes and UDAFormer: a Panorama Dataset and a Transformer-based Unsupervised Domain Adaptation Framework for Martian Terrain Segmentation[J]. IEEE Transactions on Geoscience and Remote Sensing, 2024, vol. 62, pp. 1-17, Art no. 4600117, https://doi.org/10.1109/TGRS.2023.3343109.

Comment #11:

The Ablation studies lack sufficient evidence and justification for the roles and necessity of the Identity Transformer and Refine Transformer modules.

Reviewer #3: **Summary**

The authors propose a student-teacher framework for semi-supervised segmentation of colon polyps on unlabeled video data by leveraging dynamic pseudo-label generation across teacher networks trained on multiple datasets. To measure distribution discrepancy between public labeled and private unlabeled datasets, reconstruction similarity is used a metric from public data trained auto-encoders. To enhance segmentation prediction from pseudo labels, the authors introduce refinement modules to suppress false predictions and incorporate multi-scale information to retain anatomical details. Evaluation using three public datasets for pseudo-label generation by training on a single private unlabeled dataset reveal significant gains over prior art, validating the intuition behind the designs.

**Strengths**

- The authors address a relevant task in colon polyp segmentation on unlabeled data. It is very intuitive to leverage existing labeled data to generate pseudo-labels for unlabeled samples.

- An simple and effective approach is introduced to measure distribution shifts between datasets based on generative modeling of data. Given the estimated shifts, the author leverage it as prior to enhance the learning of a student network.

- Experiments are quite extensive i.e., leveraging three public datasets provides evidence to support the claims.

**Weaknesses**

- While I appreciate the experimental setting, It is unclear if the compared methods were trained on the full public dataset (i.e., CVC-ClinicDB + CVC-ColonDB + Kvasir) or a select dataset. Since the proposed approach uses all three, it is important to explicitly highlight this point in the paper.

- As the proposed approach leverages multiple teachers, it is important to establish a lower-bound for the sensitivity of the approach by using fewer (≤ K) teachers/datasets. This can better contextualize the source of improvement since it is not clear if the approach will work with a single teacher. Moreover, the evaluation does not include popular semi-supervised approaches. (see references below).

- Yang et al. Revisiting Weak-to-Strong Consistency in Semi-Supervised Semantic Segmentation. CVPR 23

- Zhao et al. Augmentation Matters: A Simple-yet-Effective Approach to Semi-supervised

Semantic Segmentation. CVPR 23

- Chen et al. C-CAM: Causal CAM for Weakly Supervised Semantic Segmentation on Medical Image. CVPR 22

- The motivation behind using the L_tversky loss is not elaborated clearly. Additionally, the is no citation/reference to the original. More details on this objective need to be clarified.

- The writing and descriptions are not verbose. For instance, several instances of lose wording and unclear descriptions e.g., L380-381 in the selection experiment section. L400 should be ‘Evaluation metrics’.

6. PLOS authors have the option to publish the peer review history of their article (what does this mean?). If published, this will include your full peer review and any attached files.

Reviewer #1: No

Reviewer #2: **Yes: **Meibao Yao

Reviewer #3: No

---

## [Author Response · Author response to Decision Letter 0]

16 May 2024

Dear Dr. Kazunori Nagasaka,

We sincerely thank you for accepting our submission for publication with major revisions, and for giving us the opportunity to improve and resubmit our manuscript. The manuscript ID is PONE-D-24-03994. We have carefully studied each of the comments, conducted point-to-point responses, and revised the manuscript by considering all the suggestions and comments made by the reviewers.

We thank all the reviewers for appreciating our work and for their constructive suggestions. 

 We would like to thank reviewer 1 for noting that, “This paper proposed a self-adaptive teacher-student framework for polyp segmentation, to bridge the gap between the annotated datasets and the unannotated datasets. The paper is well-structured”. 

 We thank reviewer 2 for praising our method “achieving satisfactory results”. 

 We thank reviewer 3 for considering that “The authors address a relevant task in colon polyp segmentation on unlabeled data. It is very intuitive to leverage existing labeled data to generate pseudo-labels for unlabeled samples. A simple and effective approach is introduced to measure distribution shifts between datasets based on generative modeling of data. Experiments are quite extensive i.e., leveraging three public datasets provides evidence to support the claims.”

In this version, we continue to improve this paper while at the same time maintaining the merits mentioned by all the reviewers.

Please find enclosed an e-copy of the revised manuscript for further consideration and a detailed reply to all the reviewers’ comments and questions. For yours and the reviewers’ convenience, our responses shown in purple are prepared as a point-by-point response to each of the issues raised by the reviewers. Changes to the manuscript are highlighted in red.

We hope that you and the reviewers find the major revision acceptable, and we look forward to hearing from you the final decision on our submission.

Sincerely yours,

Dr. Fu Dai (on behalf of all the co-authors)

Major revisions we made include:

 We revised the manuscript thoroughly to make it concise and clearer, as well as added more details to make it comprehensive, which including: 

1) We added the explanation about teacher-student framework in medical image segmentation to clarify our method name. 

2) We added the details about the three public datasets to demonstrate our dataset is appropriate.

3) We added more explanation in each figure’s caption to make the figures easier to follow.

4) We added more details about our uncertainty-based weight to avoid future confusion. 

5) We added more details about the foreground and background attention to make the method clearer. 

 We implemented six experiments with corresponding discussion to make the experiments more comprehensive, which includes: 

1) We added four existing methods into our comparison to make the experiment more comprehensive.

2) We added an ablation study to demonstrate the necessity of our Identify Transformer Block and Refine Transformer Block.

3) We employed fewer teacher networks and fewer public datasets to validate the effectiveness of teacher networks.

– We revised Fig.4 and Fig.6 by adding two complex examples in Fig. 4, and deepening the segmentation mask color in Fig.4 and Fig.6.

All the comments and suggestions raised by the reviewers were carefully considered. Attached is our itemized response to all the points raised by the reviewers.

---

## [Decision Letter · Decision Letter 1]

17 Jun 2024

PONE-D-24-03994R1Self-Adaptive Teacher-Student Framework for Colon Polyp Segmentation from Unannotated Private Data with Public Annotated DatasetsPLOS ONE

Dear Dr. Dai,

Thank you for submitting your manuscript to PLOS ONE. After careful consideration, we feel that it has merit but does not fully meet PLOS ONE’s publication criteria as it currently stands. Therefore, we invite you to submit a revised version of the manuscript that addresses the points raised during the review process.

We look forward to receiving your revised manuscript.

Kind regards,

Kazunori Nagasaka

Academic Editor

PLOS ONE

Journal Requirements:

Additional Editor Comments:

Dear Authors,

Thank you for your submission.

As you can see in the comments, you still need some modifications regarding the manuscript.

Please consider it and we look foward to your revised manuscript.

Sincerely,

Kazunori Nagasaka

Reviewers' comments:

Reviewer's Responses to Questions

**Comments to the Author**

1. If the authors have adequately addressed your comments raised in a previous round of review and you feel that this manuscript is now acceptable for publication, you may indicate that here to bypass the “Comments to the Author” section, enter your conflict of interest statement in the “Confidential to Editor” section, and submit your "Accept" recommendation.

Reviewer #1: All comments have been addressed

Reviewer #2: All comments have been addressed

2. Is the manuscript technically sound, and do the data support the conclusions?

Reviewer #1: Yes

Reviewer #2: Yes

3. Has the statistical analysis been performed appropriately and rigorously? 

Reviewer #1: Yes

Reviewer #2: Yes

4. Have the authors made all data underlying the findings in their manuscript fully available?

Reviewer #1: Yes

Reviewer #2: Yes

5. Is the manuscript presented in an intelligible fashion and written in standard English?

Reviewer #1: Yes

Reviewer #2: Yes

6. Review Comments to the Author

Reviewer #1: Authors have addressed all the comments. I have no further comment. This paper is ready to be published.

Reviewer #2: Thank you to the authors for providing detailed responses and revisions to the previous concerns. Most of the issues in the paper have been addressed. However, there are still a few questions that need further clarification:

Comment #1:

In the experimental results section, it seems that the distribution bias among the public datasets leads to unstable prediction results. Did the authors decide the selection of the three public datasets with any specific criteria in mind, or were any pre-processing methods used to enhance the stability of pseudo-label generation?

Comment #2:

In line 176, it is mentioned that the UDFusion module uses ui to assess the reliability of pseudo-labels through contextual information. How is this contextual information represented? Is there a direct connection between formula (1) and the unstable distribution in the final visualized results?

Comment #3:

In line 189, there are two colons “:”. Is this a typographical error?

7. PLOS authors have the option to publish the peer review history of their article (what does this mean?). If published, this will include your full peer review and any attached files.

Reviewer #1: No

Reviewer #2: No

---

## [Author Response · Author response to Decision Letter 1]

28 Jun 2024

Dear Dr. Kazunori Nagasaka and reviewers:

We are very grateful for your minor revision and potentially accepting our manuscript for publication in the Journal of Plos One. We would like to thank you and all the reviewers for your very constructive comments and useful suggestions in the previous major revision, which have greatly improved our manuscript.

We have sent you a new revised version together with a letter explaining all the modifications, which contain each point raised by the reviewers, our point-by-point response, and the corresponding modifications in the manuscript. All modifications are highlighted in red in the article.

Looking forward to hearing from you about the final decision on our submission.

Sincerely Yours,

Dr. Fu Dai (on behalf of all coauthors)

---

## [Decision Letter · Decision Letter 2]

11 Jul 2024

Self-Adaptive Teacher-Student Framework for Colon Polyp Segmentation from Unannotated Private Data with Public Annotated Datasets

PONE-D-24-03994R2

Dear Dr. Dai,

We’re pleased to inform you that your manuscript has been judged scientifically suitable for publication and will be formally accepted for publication once it meets all outstanding technical requirements.

Kind regards,

Kazunori Nagasaka

Academic Editor

PLOS ONE

Additional Editor Comments (optional):

Dear Authors,

Thank you so much for your submission to Plos One.

I am pleased to inform you that we made a decision as "Accept" for your manuscript.

Your study is very useful in the field and we look forward to your future work.

Sincerely,

Kazunori Nagasaka

Reviewers' comments:

Reviewer's Responses to Questions

**Comments to the Author**

1. If the authors have adequately addressed your comments raised in a previous round of review and you feel that this manuscript is now acceptable for publication, you may indicate that here to bypass the “Comments to the Author” section, enter your conflict of interest statement in the “Confidential to Editor” section, and submit your "Accept" recommendation.

Reviewer #2: All comments have been addressed

2. Is the manuscript technically sound, and do the data support the conclusions?

Reviewer #2: Yes

3. Has the statistical analysis been performed appropriately and rigorously? 

Reviewer #2: Yes

4. Have the authors made all data underlying the findings in their manuscript fully available?

Reviewer #2: Yes

5. Is the manuscript presented in an intelligible fashion and written in standard English?

Reviewer #2: Yes

6. Review Comments to the Author

Reviewer #2: (No Response)

7. PLOS authors have the option to publish the peer review history of their article (what does this mean?). If published, this will include your full peer review and any attached files.

Reviewer #2: No

---

## [Editor Report · Acceptance letter]

16 Jul 2024

PONE-D-24-03994R2 

PLOS ONE

Dear Dr. Dai, 

I'm pleased to inform you that your manuscript has been deemed suitable for publication in PLOS ONE. Congratulations! Your manuscript is now being handed over to our production team.

Kind regards, 

on behalf of

Professor Kazunori Nagasaka 

Academic Editor

PLOS ONE